# Multi-Target Approach of *Murraya koenigii* Leaves in Treating Neurodegenerative Diseases

**DOI:** 10.3390/ph15020188

**Published:** 2022-02-02

**Authors:** Mario A. Tan, Niti Sharma, Seong Soo A. An

**Affiliations:** 1College of Science and Research Center for the Natural and Applied Sciences, University of Santo Tomas, Manila 1015, Philippines; matan@ust.edu.ph; 2Department of Bionano Technology, Gachon Bionano Research Institute, Gachon University, 1342 Seongnam-daero, Sujung-gu, Seongnam-si 461-701, Gyeonggi-do, Korea

**Keywords:** neurodegenerative disease, *Murraya koenigii*, essential oils, toxicity, carbazole alkaloids, natural products

## Abstract

Neurodegenerative diseases (NDs) mainly affect neurons and gradually lead to a loss of normal motor and cognitive functions. Atypical protein homeostasis—misfolding, aggregations and accumulations, oxidative stress, inflammation, and apoptosis—are common features in most NDs. To date, due to the complex etiology and pathogenesis of NDs, no defined treatment is available. There has been increasing interest in plant extracts as potential alternative medicines as the presence of various active components may exert synergistic and multi-pharmacological effects. *Murraya koenigii* (Rutaceae) is utilized in Ayurvedic medicine for various ailments. Pharmacological studies evidenced its potential antioxidant, anti-inflammatory, anticancer, hepatoprotective, immunomodulatory, antimicrobial, and neuroprotective activities, among others. In line with our interest in exploring natural agents for the treatment of neurodegenerative diseases, this review presents an overview of literature concerning the mechanisms of action and the safety profile of significant bioactive components present in *M. koenigii* leaves to support further investigations into their neuroprotective therapeutic potential.

## 1. Introduction

Neurodegenerative diseases (NDs) are a broad term used to define a range of disorders (Alzheimer’s disease, Parkinson’s disease, dementia with Lewy body, frontal temporal dementia, multiple sclerosis, amyotrophic lateral sclerosis, Huntington’s disease, prion disease, and many more) primarily affecting the neurons and leading, ultimately, to the progressive loss of normal motor functions and the decline in cognitive functions. The brain region affected, severity, and pace of neurodegeneration vary with the type of ND. According to a recent report, around 50 million people worldwide have such diseases [1]. The total number of Alzheimer’s disease (AD) patients is estimated to be over 100 million by the year 2050 [2]. Similarly, the number of patients with Parkinson’s disease (PD) is expected to double or more than double for populous countries such as China, India, and Indonesia [3]. 

The exact reasons for ND manifestation among some people and not for others are still not fully understood. Some NDs are suspected to occur due to genetic mutations; some are also linked to living conditions [4]. However, oxidative stress, atypical protein folding, inflammation, and apoptosis are common to the majority of NDs. Due to complex etiology and pathogenesis of NDs, to date, there are no defined treatments that can reverse the progressive degeneration of neurons and cure these diseases. Although research is progressing, drug development for NDs is slow. Over the past 30 years, only around 22 drugs were developed for NDs [5] in comparison to cancer (128 drugs), which represented the highest number, followed by antibacterial, antiviral, and antihypertension drugs (118, 110, and 79 drugs, respectively) [5]. Since many NDs are multifactorial, exploring bioactive compounds from the plants for their therapeutic potential is a safer alternative, as these bioactive compounds might act synergistically to alleviate NDs through a variety of pathways, such as preventing Aβ formation, inhibiting neurotransmission enzymes, slowing protein aggregation, removing free radicals, and reducing inflammation [6,7,8,9,10]. 

*Murraya koenigii* (curry-leaf tree) was a small, tropical to sub-tropical shrub or tree growing up to 6 m in height, belonging to the family Rutaceae. It can be found in countries like India, Sri Lanka, and Bangladesh. Almost every part (fresh leaves, fruits, bark, and roots) of this plant is used in Indian traditional system of medicine (Ayurveda) to treat various ailments. The green curry leaves were generally used in treating piles, inflammation, itching, fresh cuts, dysentery, and edema. The roots were used for curing body aches while the bark was helpful in treating snakebites [11]. Fresh *M. koenigii* leaves are known for their signature aroma, slightly spicy and bitter taste; these characteristics are preserved even after drying. Fresh and dried curry leaves are considered as an important ingredient in South Indian cuisine and are broadly used for seasoning and flavoring dishes. The curry leaves have well established therapeutic potentials like hypoglycemic, hypolipidemic [12,13,14,15], nephroprotective [16,17], hepatoprotective [18,19], gastroprotective [20], cardioprotective [21,22], atherosclerotic [23] and cholesterol lowering effects in experimental animals [24]. The alcoholic leaf extract possessed antioxidant [25,26], analgesic, anti-inflammatory [27,28,29], antipyretic [30], antitrichomonal [31], antibacterial [32,33], antifungal [34], antileishmanial [35], antidiarrheal [36], wound-healing [37], anti-obesity [38], anticancer [39,40,41], and immunomodulatory activities [42,43]. The carbazole alkaloid mahanine, purified from the curry leaves, exerted anticancer activity [44,45,46] by acting as proteasome inhibitor [47,48] and displayed antihyperglycemic action [49]. 

This review is focused on the *M. koenigii* leaf extract and its bioactive components that exhibit neuroprotective effects through various mechanisms for the prevention and treatment in NDs. The literature search was conducted using Google Scholar and PubMed online databases until November 2021 with key words being neuroprotective, *Murraya koenigii* leaf, chemical composition and essential oils. Additionally, the safety evaluation of *M. koenigii* leaves is explored.

## 2. Phytochemistry of *Murraya koenigii*


Proximate analysis on the leaves of *M. koenigii* revealed the following contents: ash (15.60% ± 0.21), moisture (23.40% ± 0.10), protein (8.38% ± 0.02), fats (6.48% ± 0.22), carbohydrate (39.44% ± 0.04), and crude fiber (6.30% ± 0.05) [50]. Phytochemical screening utilizing the same leaves of the sample identified the composition (in mg per 100 g) of various classes of secondary metabolites, including alkaloids (1.90 ± 0.01), flavonoids (7.43 ± 0.03), glycosides (0.11 ± 0.01), phenols (4.25 ± 0.04), saponins (2.50 ± 0.01), and tannins (0.86 ± 0.02) [50]. The leaves of *M. koenigii* are also a rich source of vitamins and minerals. The compositional analysis (in mg per 100 g) revealed ascorbic acid (vitamin C, 0.04 ± 0.002), β-carotene (vitamin A, 6.04 ± 0.02), niacin (vitamin B_3_, 2.73 ± 0.02), riboflavin (vitamin B_2_, 0.09 ± 0.002), thiamin (vitamin B1, 0.89 ± 0.01), and α-tocopherol (vitamin E, 0.03 ± 0.01) [50]. Among the minerals analyzed, the following composition and amount were determined (in mg per 100 g): calcium (19.73 ± 0.02), iron (0.16 ± 0.01), magnesium (49.06 ± 0.02), potassium (0.04 ± 0.001), sodium (16.50 ± 0.21), and zinc (0.04 ± 0.001) [50]. Interestingly, proximate, vitamins, and minerals analyses of the fruit pulp of *M. koenigii* revealed moisture content (64.9%), total sugar (9.76%), reducing sugar (9.58%), ascorbic acid (13.35%), calcium (0.811%), iron (0.007%), magnesium (0.166%), phosphorous (1.97%), and potassium (0.082%) [51]. 

Many structurally diverse natural products were identified from *M. koenigii*, including the leaves, roots, stem bark, fruits, and seeds, using chromatographic and spectroscopic techniques. The isolation of terpenoids, alkaloids, flavonoids, coumarins, polyphenols, and essential oils was described in phytochemical studies searching for potential biologically active constituents [52,53]. In particular, *M. koenigii* was reported as a rich source of more than fifty carbazole-type alkaloids with potent pharmacological activities [50]. Cytotoxic activity was reported for the major alkaloids koenimbine, murrayafoline, mahanine, mahanimbine, isomahanine, murrayazoline, koenoline, mahanimboline, and mukoline [52,54]. Mahanimbinine, murrayacinine, mahanimboline, isomahanine, and mahanimbine possessed diverse pharmacological activities, including antioxidant, antimicrobial, antidiabetic, and antihyperlipidemic activities [52]. Other reported antioxidant carbazole alkaloids in *M. koenigii* included mukoeic acid, murrayanine, mukonicine, koenigine, koenine, 9-formyl-3-methylcarbazole, murrayanol, and O-methylmurrayamine A, while additional antimicrobial alkaloids were girinimbine, murrayanine, mahanine, murrayacine, mukoline, and pyrafolione D [52,54]. Among the non-alkaloidal compounds identified in *M. koenigii*, the coumarins and flavonoids of catechin, epicatechin, quercetin, myricetin, naringin, and rutin were identified as antioxidant and anticancer compounds. Additionally, the polyphenolic compounds ferulic acid, gallic acid, and vanillic acid were reported to increase the antioxidant potential of *M. koenigii* [54]. 

Over 100 compounds were identified in the essential oils of *M. koenigii* leaves from different regions [26,33,55,56]. Monoterpenoids and sesquiterpenoids were reported in the leaf essential oils with therapeutic activities against various targets. Potential antioxidant and antimicrobial compounds from the essential oils of *M. koenigii* were allo-ocimene, α-terpinene, (*E*)-β-ocimene, elemol, linalool, geranyl acetate, and myrcene [54]. *M. koenigii* leaves collected from Tamilnadu, India, were extracted by hydrodistillation [26]. Gas chromatography–mass spectrometry (GC–MS) analysis of the essential oil indicated monoterpenes (11.81%), sesquiterpenes (3.12%), monoterpenoids (72.15%), and sesquiterpenoids (10.48%). Among the 33 compounds identified, allo-ocimene (5.02%), (*E*)-β-ocimene (3.68%), elemol (7.44%), geranyl acetate (6.18%), linalool (32.83%), myrcene (6.12%), α-terpinene (4.9%), and neryl acetate (3.45%) were present in the highest amounts [26]. Leaf samples collected from Vietnam were separately extracted using conventional hydrodistillation (CHD) and microwave-assisted hydro-distillation (MHD) methods. The characterization of the essential oils obtained by CHD and MHD revealed terpene hydrocarbons (90.98% and 95.79%, respectively) and oxygenated hydrocarbons (9.02% and 4.21%, respectively. In total, 76 and 62 compounds were identified using the CHD and MHD extraction methods, respectively. α-Pinene (19.03%, CHD; 29.81%, MHD), β-pinene (4.03%, CHD), β-phellandrene (18.22%, CHD; 25.62%, MHD), *trans*-β-caryophyllene (27.24%, CHD; 17.09%, MHD), β-caryophyllene (4.87%, CHD; 2.98% MHD), and bicyclogermacrene (5.23%, CHD; 3.21% MHD) were identified as the major constituents [33]. Moreover, the major components of *M. koenigii* essential oils from the London, United Kingdom were β-caryophyllene (28.7%), β-gurjunene (21.4%), selinene (8.2%), β-elemene (6.8%); essential oil from Nigeria contained oxygenated compounds including β-caryophyllene (20.5%), bicyclogermacrene (9.9%), α-cardinol (7.3%), caryophyllene epoxide (6.4%), and β-selinene (6.2%); and samples from Bangladesh contained 3-carene (54.2%) and caryophyllene (9.5%) [33]. Fresh leaves of *M. koenigii* collected from Dehra Dun, India, were hydrodistilled and subjected to GC–MS analysis. Among the 34 compounds identified, major constituents were α-pinene (51.7%), sabinene (10.5%), β-pinene (9.8%), β-caryophyllene (5.5%), limonene (5.4%), bornyl acetate (1.8%), terpinen-4-ol (1.3%), γ-terpinene (1.2%), and α-humulene (1.2%) [55]. *M. koenigii* fresh leaves were collected from six geographical locations in Malaysia and the essential oils were obtained by hydrodistillation [56]. Volatile compounds from these samples were identified as sesquiterpene hydrocarbons (36–49%), oxygenated sesquiterpenes (23–47%), oxygenated monoterpenes (6.3–35%), and oxygenated diterpenes (2.6–3.1%). Interestingly, all six essential oils identified β-caryophyllene (16.6–26.6%) and α-humulene (15.2–26.7%) as the major volatile constituents. Nine minor compounds were also identified, including aromadendrene (0.5–1.5%), caryophyllene oxide (0.7–3.6%), β-elemene (0.3–1.3%), juniper camphor (2.6–8.3%), 2-naphthalenemethanol (0.7–4.8%), β-selinene (3.8–6.5%), spathulenol (0.6–2.7%), viridiflorol (1.5–5.5%), and trivertal (0.1–1.0%) [56]. These results indicated variations in the composition of the essential oils of *M. koenigii* depending on the geographical locations and conditions including soil type and climate [56,57].

## 3. Phytochemicals of *Murraya koenigii* and Their Role in Neurodegenerative Diseases

This section describes the neuroprotective effects of *M. koenigii* leaf extracts and specified metabolites against their potential targets in NDs. The oral administration of the leaf aqueous extract (100, 200, 400 mg/kg body weight) to aluminum-treated rats was found to significantly restore antioxidant parameters (Catalase: CAT, Glutathione reduced: GSH, Lipid peroxidation: LPO), reversing them to normal through the neuroprotective properties of the extract [58]. The protective effect of *M. koenigii* on oxidative stress was most likely due to carbazole alkaloids, polyphenols, and flavonoids, which are known to have antioxidant properties [52,59,60]. Neuroprotective activity of the leaf aqueous extract (100, 200, 400 mg/kg body weight) against paraquat (10 mg/kg body weight/i.p. per week for 4 weeks)-induced Parkinsonism in rats was also studied. The leaf extract-treated animals performed better in behavioral and locomotor activities in comparison to the untreated group [61]. The possible neuroprotective potentials of a methanolic leaf extract were also observed in a two-vessel occlusion (2VO) rat model of partial global cerebral ischemia. Cognitive functions were evaluated by the Morris water maze test and the viable neurons were evaluated in the hippocampal region. The study results indicated cognitive improvement in the group treated with *M. koenigii* leaves methanolic extract [62]. Diazepam-, scopolamine- and aging-induced amnesia behavioral models in rats were evaluated for memory enhancement after treatment with *M. koenigii* leaves (MKL). MKL (leaf powder mixed with wheat flour) were orally administered to various groups, and their behaviors were evaluated with the elevated plus-maze and Hebb–Williams maze trials. A significant dose-dependent improvement in the memory scores of young and aged rats was reported, and the significant reduction in amnesia could be due to the induction by scopolamine (0.4 mg/kg, i.p.) and diazepam (1 mg/kg, i.p.) in the treatment groups [63]. In addition, the acetylcholine esterase (AChE) activity and the total cholesterol were reported to be decreased in the groups fed on the MKL diet, suggesting a role of AChE and cholesterol in inducing nootropic effects in the animals [64]. Further studies were conducted on animal models to better understand the mechanism of cognitive improvement. The 15-day treatment with *M. koenigii* alkaloid extract (MKA) caused a dose-dependent reduction in AChE levels in the brain of aged mice. It also prevented the brain from age-induced oxidative stress by decreasing lipid peroxidation and nitric oxide levels and increasing the level of antioxidant enzymes (superoxide dismutase: SOD, CAT, GSH) in the brain [65,66]. 

The neuroprotective potential of the carbazole alkaloid mahanimbine against lipopolysaccharides (LPS)-induced neuronal deficits in the SK-N-SH neuroblastoma cell line and its antioxidant potentials in the ICR mouse brain were evaluated. The pre-treatment of SK-N-SH cells with mahanimbine significantly mitigated the generation of LPS-induced reactive oxygen species (ROS) in vitro. In addition, mahanimbine also inhibited the β-secretase (IC_50_ 4 µg/mL), which was responsible for production of β amyloid (Aβ). In the in vivo study, mahanimbine supplementation re-equilibrated the levels of antioxidant markers (CAT, glutathione reductase: GR, malondialdehyde; MDA) in the brain in comparison to the untreated group [67]. Pre-treatment of LPS-challenged mice with mahanimbine enhanced the central cholinergic transmission by increasing acetylcholine (ACh) level through the AChE inhibition. Amyloidogenesis was also significantly decreased. Mahanimbine increased the levels of anti-inflammatory cytokines transforming growth factor-β (TGF-β) and interleukin-10 (IL-10), and inhibited pro-inflammatory cytokines (IL-1 β and tumor necrosis factor-α: TNF-α). Decreased activity and expression of cyclooxygenase (COX-2) gene were observed with mahanimbine supplementation in the LPS-induced group. The overall findings supported the neuroprotective potential of mahanimbine against LPS-induced neuroinflammation [68]. The anti-AChE activity (IC_50_ 0.03 mg/mL) of mahanimbine was also established in vitro under acellular conditions [69]. 

A tricyclic sesquiterpene, isolongifolene (ILF), present in the essential oil of *M. koenigii*, was studied for its neuroprotective properties both in vitro [70] and in vivo [71]. ILF displayed effective scavenging activities in various antioxidant assays with EC_50_ values of 77.34, 40.9, 16.27, 238.3, 25.01, 16.79, 1.311, 6.701, 0.418 μg/mL in the 2,2-diphenyl-1-picryl-hydrazyl-hydrate (DPPH), 2,2-azino-bis (3-ethylbenzothiazoline-6-sulfonic acid, ABTS), hydroxyl radical, nitric oxide, hydrogen peroxide, super oxide radical scavenging, ferric reducing antioxidant power (FRAP), total radical-trapping antioxidant parameter (TRAP) and reducing power assays, respectively [72]. ILF attenuated the rotenone-induced mitochondrial dysfunction and cell apoptosis in SH-SY5Y human neuroblastoma cells. ILF treatment improved the rotenone-induced oxidative stress, apoptosis, and mitochondrial dysfunction in vitro. ILF mitigated rotenone induced apoptosis by downregulating the expression of BCL2-associated X protein (Bax), caspases-3, 6, 8, and 9, cytosolic cytochrome complex (cyt c), and increasing the expression of B-cell lymphoma family of proteins (Bcl-2), along with increased mitochondrial cyt c. Additionally, the regulation of the expression of phosphoinositide 3-kinase (p-PI3K), protein kinase B (p-AKT), and glycogen synthase kinase 3β (p-GSK3β) by ILF demonstrated its neuroprotective effects [70]. The PI3K/AKT/GSK3β signaling pathway is an important pathway for neuronal growth and function [73]. AKT regulated PI3K through the phosphorylation of GSK3β, which was identified to be associated with various vital cellular functions, such as the regulation of cell apoptosis and survival [74]. The active form of GSK-3β (p-GSK-3 Tyr216) was reported to be increased in postmortem striata of PD patients [75]. These events occur through mitochondrially mediated cell death pathways [76], which involve the activation and localization of Bcl-2 and mitochondrial complex I activity [77]. In a separate study [71], ILF improved the oxidative stress condition and movement impairment induced by rotenone in a rat PD model. The co-administration of ILF and rotenone significantly restored the levels of antioxidant enzymes (SOD, CAT, and glutathione peroxidase (GPx)) and decreased the level of LPO. In addition, ILF improved the motor dysfunction in muscles and catalepsy that were induced by rotenone. This suggested that ILF could protect the dopaminergic neurons through free radical scavenging activity [70,71]. 

A sesquiterpene alkene and atypical dietary cannabinoid, β-caryophyllene (BCP), is another essential bioactive compound in *M. koenigii* leaves [78]. It is a selective agonist of cannabinoid receptor 2 (CB2R), making it a favorable target for immune modulation, inflammation, and neuropathic pain conditions [79], and is involved in pathogenesis of cancer and NDs [80,81]. Through the CBR signaling pathways, the endocannabinoid system (ECS) controls stress linked to cognitive and emotional responses. The dysregulation of the ECS plays an important role in the pathogenesis and progression of various NDs, such as multiple sclerosis (MS), amyotrophic lateral sclerosis (ALS), and Alzheimer’s disease (AD) [82,83]. In the brain of MS patients, aggregates of CB1R- and CB2R-expressing cells were found in the active multiple sclerosis plaques [84]. Similarly, in case of ALS patients, the increased CB2R-positive microglia and macrophages were present in the areas of motor neuron damage [85]. In AD, the altered expressions of CB1R and CB2R were observed in the brain [86,87,88]. In addition, increased levels of endocannabinoid degradation enzymes namely, fatty acid amide hydrolase (FAAH) and monoacylglycerol lipase (MAGL), were found in human AD brains [87,89]. Hence, reversing ECS dysregulation through the activation of CB2R and the neuroprotection by BCP may influence the regulation of myelination and stimulating immune balance along with the inhibition of endocannabinoid degradation enzymes for their positive effects in NDs [90]. 

The effect of BCP on cognition was studied in a galactose (GAL) model of aging in mice. GAL induction lead to increased DNA oxidation in the prefrontal cortex and increased astrocyte numbers and interactions in the hippocampal region of the brain. Although the cognitive damage was not reversed by administering BCP, it was able to block the increases in the number of astrocytes and DNA oxidation in mice treated with GAL. Hence, the neuroprotective effects of BCP were demonstrated at the molecular and cellular level in the GAL model of aging [91]. The protective effect of BCP was also seen in isolated macrophages and lymphocytes and in vivo models of experimental autoimmune encephalomyelitis (EAE) [92]. The in vitro neuroprotective effect of BCP was reported against the LPS-induced oligodendrocyte toxicity, which was facilitated through the nuclear factor-erythroid 2-related factor 2 and heme oxygenase-1 (Nrf2/HO-1) antioxidant defense pathway and peroxisome proliferator-activated receptor gamma (PPAR-γ) signaling pathways through CB2R binding [93]. Similarly, BCP (48 mg/kg) alleviated the Alzheimer-like diseases by modulating CB2R and PPAR-γ signaling pathways in another study [94]. At 100 μM, BCP inhibited the expression of inducible nitric oxide synthetase (iNOS), interleukin (IL)-1β, IL-6, and COX2 in C6 microglial cells, and decreased the level of nitric oxide and prostaglandin E2. These studies show that BCP had good neuroprotective potential, which may be related to the modulation of inflammatory mediators [95] in association with high mobility group box 1 and toll-like receptor 4 (HMGB1/TLR4) signaling [96], reducing oxidative stress and modulating CB2R [97,98,99]. 

Spathulenol (5,10-cycloaromadendrane), a tricyclic sesquiterpenoid, is a component of essential oil from the leaves of *M. koenigii* [100]. Spathulenol exerted neuroprotective effects against 6-hydroxydopamine (6-OHDA), a synthetic neurotoxin, in SH-SY5Y human dopaminergic neuroblastoma cells. Spathulenol not only relieved the generated oxidative stress in the cells due to 6-OHDG, but the mitochondrial membrane integrity was maintained, highlighting its potentials as a promising therapy for treating NDs [101]. 

(+)-3-Carene, a bicyclic monoterpenoid from *M. koenigii* leaf [78], was reported to be a potent uncompetitive inhibitor of AChE (IC_50_ 0.2 mM) [102]. A molecular docking study envisaging ligand–target interactions of essential oil components with human AChE, led to seven hit compounds, which formed favorable bonds at the enzyme active site. Hence, the in vitro evaluation led to the discovery of two new active hits, namely caryophyllene oxide (IC_50_ 5.3 µM) and geranyl acetate (IC_50_ 244.5 μM) [103], as components of essential oil from *M. koenigii* leaf [26,78]. 

Linalool, a monoterpene alcohol, was also present in the essential oil of *M. koenigii* leaves. The improved cognitive-enhancing effects and anti-apoptotic activities in Aβ1-42-treated rats could mostly result from its antioxidant potential [104]. In another study, linalool displayed neuroprotective potential in male Wistar rats after acrylamide (50 mg/kg i.p.) treatment. Acrylamide was found in various food products during processing, which could present a potent neurotoxic effect in human and animals [105]. Linalool treatment (12.5 mg/kg i.p.) significantly reduced movement, which could be due to the acrylamide-induced abnormalities. Increased GSH levels in the rat brain could be regulated by linalool, indicating its role of antioxidant properties in neuroprotection [106]. In addition to improving cognitive functions, linalool displayed a neuroprotective effect through competitively inhibiting glutamate release and non-competitively blocking of N-methyl-D-aspartate (NMDA) receptors [107,108]. Moreover, it also decreased microgliosis and astrogliosis, reduced the formation of neurofibrillary tangles and β-amyloid plaques, and the levels of pro-inflammatory markers in the brain [109].

The neuroprotective effects of β-myrcene (MYR), a monoterpene present in the leaves of *M. koenigii*, were studied. MYR treatment protected the C57BL/J6 mice against oxidative stress, apoptosis, and histopathological damage induced by cerebral ischemia/reperfusion (I/R) [110]. 

The neuroprotective effect of nerolidol (NRD), a sesquiterpene alcohol, was discovered in the essential oil of *M. koenigii* leaves from the studies with a rotenone-induced model of PD [100]. NRD supplementation considerably improved the levels of oxidative stress markers (SOD, CAT, GSH, and MDA). NRD also inhibited the release of pro-inflammatory cytokines and inflammatory mediators. Moreover, NRD treatment prevented the rotenone-induced activation of glial cells and the loss of dopaminergic neurons and nerve fibers, thus mitigating rotenone-induced dopaminergic neurodegeneration [111]. In another study, NRD significantly improved the locomotor activities and cognitive impairment, and reduced the AChE activity and reduction in oxidative stress [112]. Therefore, NRD displays its neuroprotective potential by antioxidant and anti-inflammatory action, which would be helpful in treating NDs. 

Table 1 summarized the mechanism of action of different extracts and bioactive components in ameliorating the symptoms of NDs. Structures of the biologically active natural products of *M. koenigii* against various NDs are shown in Figure 1. A summative representation of the mechanism of action of the *M. koenigii* extracts and natural products are shown in Figure 2.

## 4. Clinical Studies

Currently, no significant clinical trials have been conducted on curry leaves. A randomized controlled clinical trial was conducted to evaluate the efficacy of *M. koenigii* and chlorhexidine gluconate for the treatment of gingivitis [113]. In another study, curry leaf powder (5 g daily for 45 days) was given to volunteers to evaluate its effect on liver and renal functions, and no harmful effect was observed on the functions of both the liver and kidney [114]. An herbal medicine containing curry leaves along with pomegranate and turmeric was developed for the treatment of irritable bowel syndrome patients [115]. The effect of curry leaves was also investigated in hypertensive subjects [116]. However, despite numerous in vivo and in vitro studies exploring the neuroprotective potential of *M. koenigii*, the research was lacking on its preclinical and/or clinical efficacy. Therefore, there is an urgent need to conduct clinical trials to prove the neuroprotective capacity of *M. koenigii*.

## 5. Toxicity of *Murraya koenigii* Leaves

Curry leaves were rich in calcium, potassium, magnesium, and phosphorous, along with trace amounts of zinc, manganese, selenium, and iron. Additionally, lead, mercury, and cadmium were also present, but below the US FDA limits [117]. Various leaf samples of *M. koenigii* were collected from different regions, and their elemental analyses were performed by the instrumental neutron activation (INAA) and atomic absorption spectrophotometry (AAS). Mean elemental contents varied owing to different climate, soil, and other geo-environmental conditions. The changes in plants grown on polluted soils revealed the higher levels of heavy metals, which posed a major health concern, where they could affect kidney, liver, and the central nervous system [118].

No signs of mortality or morbidity were seen in either male or female rats fed the ethanolic extract of *M. koenigii* leaf (300 and 500 mg/kg) for 28 days. In addition, no mortality was observed at higher doses (900 mg/kg), but congestion, hemorrhage, and lymphocyte infiltration were recorded. The study concluded that the safety consumption could be high as 500 mg/kg dosage without inducing any structural damage to the organs [118]. Similarly, no mortality or toxicity signs were observed in another toxicological study of Malaysian MKL. For MKL methanolic extract an LD_50_ = 200 mg/kg/day was reported [119]. In addition, mice fed with the mahanine-enriched fraction (MEF) at different doses (5000 mg/kg BW single dose, 300–1500 mg/kg BW/day for 14 days, and 300 mg/kg BW for 180 days) did not exhibit significant toxicity, mortality, or behavioral changes [120]. Another study suggested that crude leaf powder and methanolic leaf extract was safe up to 9000 mg/kg in mice [121] However, methanolic leaf extract was found to be moderately toxic (LD_50_ = 316.23 mg/kg body weight) to rats and to cause liver inflammation. However, this dose had no severe side effects on other organs [31]. 

## 6. Conclusions and Future Directions

Traditional medicinal plants have gained importance during the last few decades, and evidence has been presented for their therapeutic actions, allowing the traditional knowledge of plants to be streamlined with the modern system of medicine to achieve health benefits. This review presented substantial evidence in relation to the mechanistic aspects of *M. koenigii* leaves in ameliorating several neurodegenerative disorders. *M. koenigii* leaves are an integral part of Indian cuisine and are used for various ailments traditionally. Carbazole alkaloids are abundantly found in species of the Rutaceae family including *M. koenigii* (Curry leaves). Both the natural and synthetic derivatives of carbazole alkaloids revealed numerous pharmacological activities, including neuroprotection. The *M. koenigii* leaf extracts and their bioactive compounds, including carbazole alkaloids, sesquiterpenoids, and monoterpenoids, exhibited a multi-target approach by alleviating the oxidative stress, attenuating proinflammatory cytokines, inhibiting AChE and BACE 1, preventing/reducing Aβ protein aggregation, and improving cognitive dysfunction. Thus, *M. koenigii* leaves, their extracts, or purified compounds could offer a beneficial alternative therapy to treat NDs through their multi-targeted neuroprotective properties and by improving cholinergic transmissions. Hence, the bioactive compounds from the leaves can serve as lead molecules in future drug discovery. Moreover, there have been no safety concerns associated with the use of *M. koenigii* leaves. Therefore, it is highly recommended to determine effective dose of leaves or their compounds in human for future clinical trials on NDs.

## Figures and Tables

**Figure 1 pharmaceuticals-15-00188-f001:**
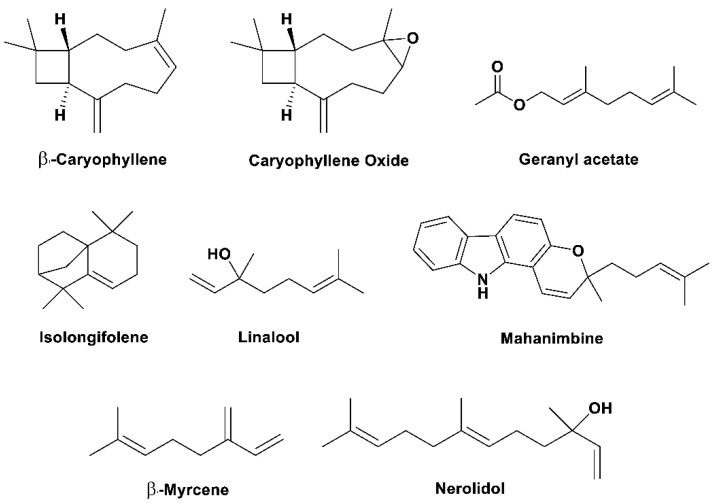
Biologically active natural products from *Murraya koenigii* against neurodegenerative diseases.

**Figure 2 pharmaceuticals-15-00188-f002:**
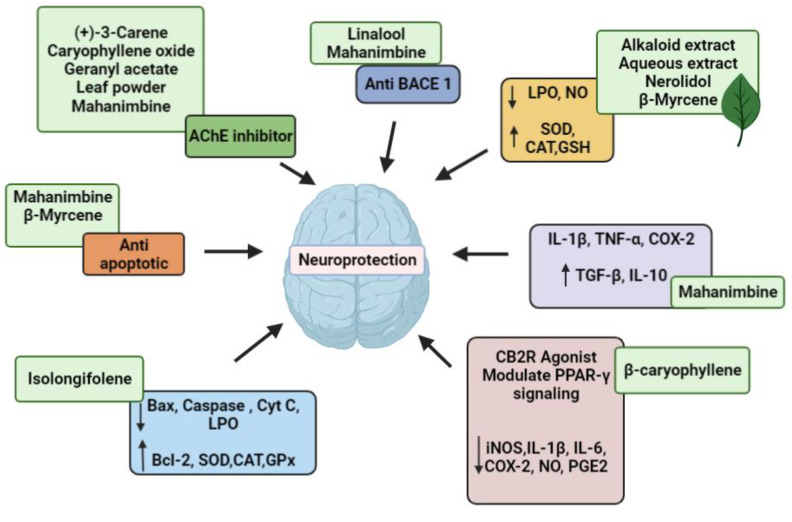
Mechanisms of action of the identified bioactive compounds from *Murraya koenigii*.

**Table 1 pharmaceuticals-15-00188-t001:** Mechanism of action of different extracts of *Murraya koeingii* leaves and their bioactive components in ameliorating symptoms of NDs.

Extract/Bioactive Components	Model	Property	Neuroprotective Mechanism	Refs.
Alkaloid extract	Swiss Albino aged mice (in vivo)	Antioxidant properties	Reduced LPO, NO, Increased SOD, CAT, GSH	[65,66]
Aqueous extract	Aluminum-treated rats (in vivo)	Antioxidant properties	Decreased LPO, Increased CAT, GSH	[58]
Aqueous extract	Paraquat-induced Parkinsonism in rats (in vivo)	Antioxidant properties	Better performance in behavioral and locomotor activities, Increased CAT, GSH, Decreased LPO	[61]
β-Caryophyllene	Brain of patients (Postmortum) Galactose model of aging mice (in vivo) Autoimmune encephalomyelitis model (in vivo) Isolated macrophages and lymphocytes (in vitro)	Agonist of CB2R Antioxidant properties, Anti-inflammatory	Activation of CB2R, Block the increases in the number of astrocytes and the DNA oxidation, Modulated CB2R and PPAR-γ signaling pathways, Inhibited expression of iNOS, IL-1β, IL-6, and COX-2 and decreased the level of nitric oxide and prostaglandin E_2_	[84,86,87,88,91,92,94]
(+)-3-Carene	In vitro	Enzyme inhibition	Uncompetitive inhibitor of AChE	[102]
Caryophyllene oxide	In vitro, in silico	Enzyme inhibition	Inhibitor of AChE	[103]
Geranyl acetate	In vitro, in silico	Enzyme inhibition	Inhibitor of AChE	[103]
Isolongifolene	SH-SY5Y (in vitro) Rotenone-induced rat model of PD (in vivo)	Antioxidant properties Protect dopaminergic neurons	Downregulated the expression of Bax, caspases-3, 6, 8, and 9, cytosolic cyt c; increased Bcl-2 expression, Increased SOD, CAT, GPx, Decreased LPO	[70,71]
Leaf powder	Diazepam-, scopolamine- and ageing-induced amnesia behavioral models in rats (in vivo)	Nootropic effect pro-cholinergic activity	Improved memory and learning impairment, Decreased AChE and total cholesterol levels	[63,64]
Linalool	Aβ1-42-treated rats (in vivo)	Antioxidant properties	Cognitive-enhancing effects, anti-apoptotic activities, NMDA receptor antagonist	[104,106,107]
Mahanimbine	SK-N-SH (in vitro) ICR mouse (in vivo)	Antioxidant properties, Anti-inflammatory	Inhibited BACE1 and AChE, Decreased IL-1 β and TNF-α, COX2, Increased TGF-β and IL-10	[67,69]
Methanolic extract	Two-vessel occlusion rat model of partial global cerebral ischemia (in vivo)	Nootropic effect	Improved memory and learning impairment	[62]
β-Myrcene	Global cerebral ischemia/reperfusion (I/R) in C57BL/J6 mice	Antioxidant properties	Protection against oxidative stress, apoptosis, and histopathological damage	[107]
Nerolidol	Rotenone-induced model of PD (in vivo)	Antioxidant properties, Anti-inflammatory	Increased SOD, CAT, GSH, Decreased MDA inhibits the release of pro-inflammatory cytokines and inflammatory mediators, Prevented rotenone-induced activation of glial cells, Improved locomotor activity and cognitive impairment, Reduced the AChE activity	[100,111,112]
Spathulenol	SH-SY5Y (in vitro)	Antioxidant properties	Maintained mitochondrial membrane integrity	[101]

## Data Availability

Data sharing not applicable.

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
