# Peer review of "Multi-Target Approach of Murraya koenigii Leaves in Treating Neurodegenerative Diseases"

_pharmaceuticals, 2022, doi:10.3390/ph15020188_

Round 1
Reviewer 1 Report
The manuscript presents the overview of literature concerning mechanisms of action and safety profile of significant bioactive components present in M. koenigii leaves, what is important for the further investigations on their neuroprotective therapeutic potentials. It is very interesting and comprehensive, with great number of references. The presented literature overview is thoroughly discussed with important author` s statements and highlights, especially regarding the molecular mechanism of action.
According to my opinion the manuscript should be accepted for the publication in Pharmaceuticals.
Minor comments are as follow:
I propose to replace the Ref. 5 with more appropriate one.
In line 80: moisture content 23,4 % +- 0,1 (the same number of figures for mean and standard deviation)
Author Response
Reviewer 1
Comments and Suggestions for Authors
The manuscript presents the overview of literature concerning mechanisms of action and safety profile of significant bioactive components present in M. koenigii leaves, what is important for the further investigations on their neuroprotective therapeutic potentials. It is very interesting and comprehensive, with great number of references. The presented literature overview is thoroughly discussed with important author` s statements and highlights, especially regarding the molecular mechanism of action.
According to my opinion the manuscript should be accepted for the publication in Pharmaceuticals.
We wish to express our thanks to the reviewer for the inputs and suggestions on our manuscript. We have incorporated the changes in the manuscript in red fonts.
Minor comments are as follow:
- I propose to replace the Ref. 5 with more appropriate one.
√ We have replaced Ref. 5 and added more appropriate reference as suggested by the reviewer.
- In line 80: moisture content 23,4 % +- 0,1 (the same number of figures for mean and standard deviation)
√ We have changed 23.4 to 23.40, and 0.1 to 0.10 to be consistent with significant number of figures with the other values.
Reviewer 2 Report
The manuscript “Multi-target approach of Murraya koenigii leaves in treating neurodegenerative diseases” is very interesting. Since the incidence in neurodegenerative diseases is increasing new approach to their treatment is valuable. The manuscript is well written and conceptualized. Introduction is clear and concise, and conclusion is relevant. Only minor changes are needed.
Minor comments
Page 2, line 47. The authors could give the reference (the source) for the number of drugs that they are mentioning.
Page 2, line 62. Full stop should be omitted.
Page 3, lines 120-121, and 141-142. Abbreviation GC-MS should be explained earlier.
Page 4, line 175. It is only stated “M. koenigii leaves” but is not stated how extract is prepared.
Page 4, line 184. Again, maybe the extract preparation should be stated.
Page 4, line 185. The AChE abbreviation is already introduced.
Page 5, line 205. “… was also established in vitro”. On which cell model?
Page 5, line 227. Abbreviation for Parkinson’s disease is already introduced.
Page 10, line 364. Add “=” between LD50 and the number.
Author Response
Reviewer 2 Comments and Suggestions for Authors
The manuscript “Multi-target approach of Murraya koenigii leaves in treating neurodegenerative diseases” is very interesting. Since the incidence in neurodegenerative diseases is increasing new approach to their treatment is valuable. The manuscript is well written and conceptualized. Introduction is clear and concise, and conclusion is relevant. Only minor changes are needed.
Our sincere thanks to the reviewer for the comments and suggestions on the improvement of our manuscript. All changes in the manuscript are indicated in red fonts.
- Page 2, line 47. The authors could give the reference (the source) for the number of drugs that they are mentioning.
√ The reference has been cited as suggested by the reviewer.
- Page 2, line 62. Full stop should be omitted.
√ The Full stop has been omitted.
- Page 3, lines 120-121, and 141-142. Abbreviation GC-MS should be explained earlier.
√ The changes have been made as suggested.
- Page 4, line 175. It is only stated “ koenigiileaves” but is not stated how extract is prepared.
√ The information has been provided.
- Page 4, line 184. Again, maybe the extract preparation should be stated.
√ The information has been provided.
- Page 4, line 185. The AChE abbreviation is already introduced.
√ The suggested corrections have been made.
- Page 5, line 205. “… was also established in vitro”. On which cell model?
√ The information has been provided.
- Page 5, line 227. Abbreviation for Parkinson’s disease is already introduced.
√ The suggested corrections have been made.
- Page 10, line 364. Add “=” between LD50 and the number.
√ The suggested corrections have been made

Reviewer 3 Report
In this review article, the authors described the neuroprotective effects of Murraya koenigii (Rutaceae), a small, tropical to sub-tropical plant. In detail, the review was focused on the M. koenigii leaf extract and its bioactive components. The review is divided into two main parts: i) the phytochemistry of Murraya koenigii; ii) the biological and toxicity aspects. Overall, the review is well-written and the figures and tables are clear to understand.
I have only some minor comments:
- Line 336. ...immense neuroprotective capacity... immense?
- Conclusions. I suggest highlighting the benefits of M. koenigii leaves compared to other neuroprotective compounds derived from plants.
Author Response
Reviewer 3
Comments and Suggestions for Authors
In this review article, the authors described the neuroprotective effects of Murraya koenigii (Rutaceae), a small, tropical to sub-tropical plant. In detail, the review was focused on the M. koenigii leaf extract and its bioactive components. The review is divided into two main parts: i) the phytochemistry of Murraya koenigii; ii) the biological and toxicity aspects. Overall, the review is well-written and the figures and tables are clear to understand.
We are grateful to the reviewer for his valuable suggestions on our manuscript. All changes in the manuscript are indicated in red fonts.
I have only some minor comments:
- Line 336. ...immense neuroprotective capacity... immense?
√ The word ‘immense’ has been removed from the sentence.
- I suggest highlighting the benefits of M. koenigii leaves compared to other neuroprotective compounds derived from plants.
√ We have incorporated additional information highlighting the benefits of M. koenigii in the conclusion part as suggested by the reviewer.
